# IMPROVING GUI GROUNDING WITH EXPLICIT POSITION-TO-COORDINATE MAPPING

## ABSTRACT

GUI grounding, the task of mapping natural-language instructions to pixel coordinates, is crucial for autonomous agents, yet remains difficult for current VLMs. The core bottleneck is reliable patch-to-pixel mapping, which breaks when extrapolating to high-resolution displays unseen during training. Current approaches generate coordinates as text tokens directly from visual features, forcing the model to infer complex position-to-pixel mappings implicitly; as a result, accuracy degrades and failures proliferate on new resolutions. We address this with two complementary innovations. First, **RULER tokens** serve as explicit coordinate markers, letting the model reference positions similar to gridlines on a map and *adjust* rather than generate coordinates from scratch. Second, **Interleaved MRoPE (I-MRoPE)** improves spatial encoding by ensuring that width and height dimensions are represented equally, addressing the asymmetry of standard positional schemes. Experiments on ScreenSpot, ScreenSpot-V2, and ScreenSpot-Pro show consistent gains in grounding accuracy, with the largest improvements on high-resolution interfaces. By providing explicit spatial guidance rather than relying on implicit learning, our approach enables more reliable GUI automation across diverse resolutions and platforms.[1]

## 1 INTRODUCTION

GUI grounding is the task of mapping natural language instructions to precise pixel coordinates in graphical user interfaces, enabling autonomous agents to interact with software as humans do (Zhang et al., 2025a; Wang et al., 2024a; Zheng et al., 2024). This capability is fundamental for computer automation: without accurate grounding, agents cannot click buttons, fill forms, or navigate interfaces reliably. Although early approaches relied on structured metadata from HTML/DOM trees or accessibility APIs (Li et al., 2020; Deng et al., 2023), these methods face significant limitations: they require access to the underlying UI structure, which is often unavailable in desktop applications, inconsistent across platforms, or completely absent in legacy systems. Pure vision-based grounding, which operates directly on screenshots, offers universal applicability across any visual interface without requiring special access or instrumentation (Qin et al., 2025; Wang et al., 2025b; Guo et al., 2025). This approach mirrors human interaction with GUIs and enables automation of any software visible on screen, from modern web applications to legacy desktop tools.

Current vision-based approaches typically formulate GUI grounding as a coordinate generation task, where models output pixel positions as text tokens (e.g., "x=523, y=217"). This paradigm, adopted by models such as SeeClick (Cheng et al., 2024), CogAgent (Hong et al., 2024), and UI-TARS (Qin et al., 2025), treats coordinate prediction as a standard language modeling problem. However, this approach faces a fundamental challenge illustrated in Figure 1: models must learn to map from high-dimensional visual positional embeddings to precise numerical coordinates as token outputs without explicit spatial guidance. The mapping is entirely *implicit*: the model receives visual patches with positional embeddings and must learn to translate these abstract and similar representations into exact and distinct pixel value tokens through its language modeling head.

This implicit approach leads to two critical problems. First, **unreliable coordinate prediction**: Without explicit guidance linking positions to coordinates, models struggle to learn stable mappings,

---

[1]Our code and models will be public upon acceptance.

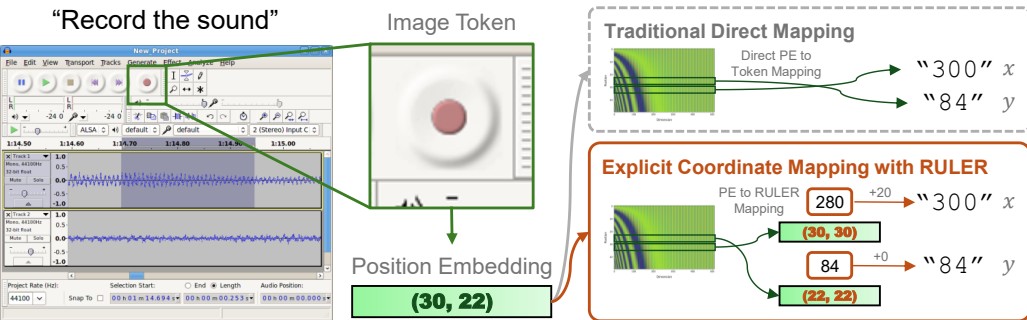

Figure 1: A comparison between traditional direct positional embedding-to-pixel coordinate mapping and RULER's explicit coordinate mapping.

requiring extensive training data and still producing inconsistent results (Gou et al., 2025; Wu et al., 2025a). Second, **poor resolution generalization**: Models trained on specific resolutions generally fail when deployed on different screen sizes, as the implicit mapping function learned during training does not transfer to new coordinate ranges (Nayak et al., 2025; Li et al., 2025b).

We also identify a technical limitation in the way current VLMs encode spatial information. Standard Multidimensional Rotary Positional Embedding (MRoPE), used in state-of-the-art models like Qwen2-VL and Qwen2.5-VL (Wang et al., 2024b; Bai et al., 2025), assigns different frequency bands to height and width dimensions sequentially. This creates an imbalance where one dimension receives only high-frequency components while another receives only low-frequency components, leading to uneven spatial modeling capabilities across axes, a previously overlooked issue that impacts grounding precision.

To address these challenges, we introduce a framework that provides explicit spatial guidance for GUI grounding through two key innovations:

Firstly, **RULER (Rotary position-to-pixeL mappER) tokens** establish an explicit coordinate reference system within the model. As illustrated in Figure 1, these auxiliary tokens encode pixel coordinates directly and share positional embeddings with the corresponding image patches. Instead of regressing the coordinates from abstract features, the models can now refer to the nearest RULER token and perform simple bounded arithmetic to determine exact positions. This transforms an unstable regression problem into a robust reference-and-adjustment mechanism, similar to how humans might use gridlines on a map.

Secondly, **Interleaved MRoPE (I-MRoPE)** addresses frequency imbalance in standard positional encodings. By interleaving rather than sequentially assigning frequency components across spatial dimensions, it distributes high- and low-frequency signals uniformly across width and height. This produces balanced spatial representations and improves the model's ability to distinguish positions along both axes equally.

Training models from scratch with our framework and finetuning existing VLMs with RULER tokens, we perform extensive evaluation on ScreenSpot (Cheng et al., 2024), ScreenSpot-V2 (Wu et al., 2025b), and ScreenSpot-Pro (Li et al., 2025b). Our approach achieves significant improvements: on the challenging ScreenSpot Pro benchmark with high-resolution displays exceeding our training resolution, we improve accuracy from 31.1% to 37.2% through finetuning alone, demonstrating strong generalization capability. These gains are achieved with minimal computational overhead, as RULER tokens add less than 1% to the total token count even for 8K displays.

Our work makes three key contributions: (1) We identify and formalize the implicit mapping problem in current GUI grounding approaches, showing how it leads to poor accuracy and resolution brittleness; (2) We introduce RULER tokens, an explicit coordinate reference mechanism that transforms unstable regression into robust spatial referencing; (3) We present I-MRoPE, a balanced positional embedding scheme that provides equal spatial modeling capacity across dimensions. Together, these innovations establish a more principled approach to GUI grounding that treats pixel-level precision as an explicit architectural concern rather than an emergent property.

## 2 RELATED WORK

**Positional Embeddings in Vision-Languge Models.**   Rotary Positional Embedding (RoPE) (Su et al., 2024) encodes positions by rotating embedding dimension pairs with angles proportional to token indices, but suffers from a long-term decay bias in low-frequency components. HoPE (Li et al., 2025a) zeros out these low-frequency terms to prevent long-range bias. For vision-language models, abundant visual tokens exhaust RoPE's context window; V2PE (Ge et al., 2024) rescales step sizes for vision tokens, while CircleRoPE (Wang et al., 2025a) projects image tokens into circular space orthogonal to text, ensuring equal cross-modal distances. For video, M-RoPE (Wang et al., 2024b) separately encodes spatial-temporal dimensions but disrupts cross-modal alignment by offsetting text tokens. Video RoPE (Liu et al., 2025) addresses this by rotating spatial positions while preserving text-video continuity and relative spatial information. Currently, Qwen2-VL and Qwen2.5-VL's MRoPE (Wang et al., 2024b; Bai et al., 2025) is one of the most prevailing multidimensional positional embedding due to the popularity of these models. However, the implementation of MRoPE results in a biased partition of RoPE features for each spatial-temporal dimensions. Our I-MROPE provides an elegant improvement to MRoPE that provides a full frequency spectrum of RoPE features for each spatial-temporal dimension, which allows the model to perform better position perception.

**GUI Grounding Models.**   Given the limitations of general-purpose models on UI grounding tasks (Li et al., 2025b; Nayak et al., 2025), recent work has focused on developing task-specific models. Early approaches formulated coordinate prediction (UI grounding) as a text generation problem. For example, JEDI (Xie et al., 2025) and UI-TARS (Qin et al., 2025) finetune open-source VLMs on synthetically generated data to enhance grounding capabilities. Building on this, GTA1 (Yang et al., 2025) and SE-GUI (Yuan et al., 2025) leverage reinforcement learning, specifically GRPO (Shao et al., 2024), with rule-based rewards to self-improve grounding performance. PHI-GROUND (Zhang et al., 2025b) introduces a label smoothing strategy that weights coordinate token predictions by their numerical distance from the ground truth, while emphasizing digit positions (e.g., tens, hundreds). In contrast, some recent approaches have moved away from text-based coordinate generation. For example, GUI-ACTOR (Wu et al., 2025a) proposes coordinate-free grounding, where the model directly predicts the visual patches corresponding to the target locations. However, current methods either generate coordinates as natural language response, which requires mapping positional embeddings to number tokens, or requires large changes to the model architecture, which is not directly compatible with general tasks. Our introduced RULER provides both explicit guidance for mapping position information to tokens, while keeping the model's original autoregressive generation design to maximize compatibility with other model usage scenarios.

## 3 METHOD

We present a framework for UI grounding that addresses fundamental limitations in how current VLMs handle spatial perception. Our approach introduces two complementary innovations: **(i) Interleaved Multidimensional Rotary Positional Embedding (I-MROPE)** that provides balanced spatial representations, and **(ii) RULER** tokens that establish explicit position-to-pixel coordinate mappings. We provide an overview of our proposed method in Figure 2.

### 3.1 RULER: EXPLICIT POSITION-TO-PIXEL COORDINATE MAPPING

Current VLMs predict pixel coordinates for GUI grounding by generating coordinates as text tokens (e.g., "x=523, y=217"). Since the source of such coordinate-related information is only recorded by image tokens' positional embeddings, generating coordinate tokens requires implicit and direct mapping from high-dimensional visual features' positional embeddings to natural language number tokens. This approach suffers from unstable learning dynamics and poor generalization to unseen resolutions, as the learned regression functions are inherently resolution-specific (Gou et al., 2025; Wu et al., 2025a).

To provide a more explicit guidance for the model in generating pixel coordinates, we propose RULER, which introduces auxiliary tokens that explicitly encode pixel coordinates and share positional embeddings with corresponding image patches. Inspired by the induction head mechanism in

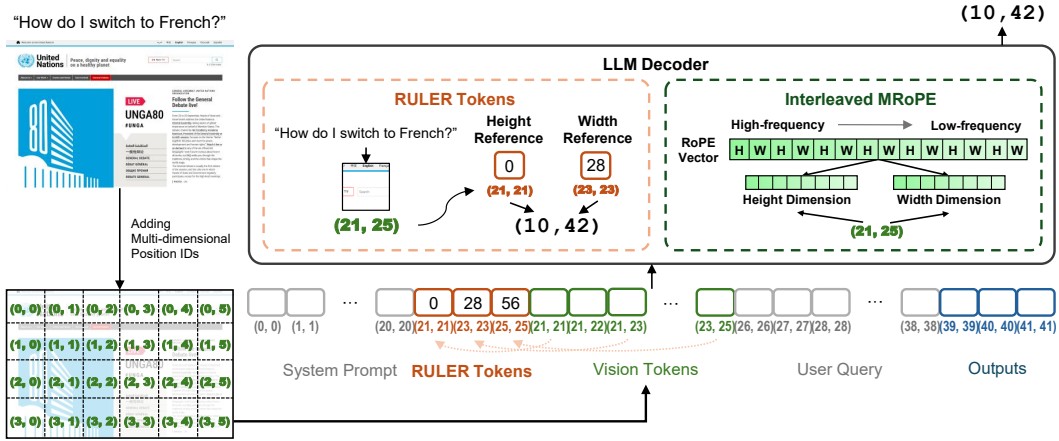

Figure 2: **Model architecture.** Our framework augments vision-language models with two key innovations: (1) RULER tokens that provide explicit position-to-coordinate mappings, transforming coordinate prediction from regression to retrieval, and (2) I-MRoPE that rebalances positional embeddings by interleaving frequency components across spatial dimensions, ensuring equal representational capacity for width and height, and

pretrained Transformers (Olsson et al., 2022), we take advantage of the model's learned capability to compare position IDs and to copy tokens according to their positions, and use a series of tokens with carefully designed position IDs and token values as a ruler for the image. With the help of these tokens, instead of regressing pixel values from positional embeddings, the model finds a RULER token whose positional encoding best aligns with an image patch, and copy its value as a reference coordinate value. Based on the retrieved coordinate value, the model only needs to add a number bounded by a constant $b$ internally to get the final output number, where $b$ is irrelevant of the image resolution, reducing the generalization gap on images with higher resolutions than the trained ones. An illustrated comparison between RULER and traditional grounding methods is shown in Figure 1.

Specifically, consider an image partitioned (tokenized) into $H \times W$ patches each covering $p \times p$ pixels, and let $\mathbf{x}_{\text{sys}}$ denote system tokens, $\mathbf{x}_{\text{vision}}$ the visual patch embeddings, and $\mathbf{x}_{\text{prompt}}$ the text prompt embeddings. We augment the input sequence with a set of auxiliary coordinate tokens $\mathbf{x}_{\text{RULER}}$ as follows:

$$\mathbf{x}_{\text{input}} = \left[ \mathbf{x}_{\text{sys}}, \mathbf{x}_{\text{RULER}}, \mathbf{x}_{\text{vision}}, \mathbf{x}_{\text{prompt}} \right], \tag{1}$$

We construct each RULER token $r_i \in \mathbf{x}_{\text{RULER}}$ so that it shares the same spatial position ID as a visual patch and has the face token value of the initial pixel coordinate of the corresponding visual patch. This construction both aligns RULER's position with input visual patches and aligns its value with output coordinate tokens; thus, bridges the position-to-coordinate mapping:

$$\text{PE}_{\text{RULER}}(r_i) = \mathbf{R}^{\text{MRoPE}}_{\Theta, t_0 + i} \tag{2}$$

where $\mathbf{R}^{\text{MRoPE}}$ is a multidimensional RoPE operator, and $t_0$ is a fixed temporal index ensuring that the height and width components match those of the vision token at spatial position $i$. In practice, $t_0$ is the initial spatial position ID of the image patches. Note that RULER only models one of the multiple dimensions of spatial position IDs, since $t_0$ is the same for both height and width dimensions, and each image patch covers a square part of image. Thus, the mapping between height or width to the pixel coordinate values is identical. This sharing of RULER mapping on multiple spatial dimensions helps reduce the number of RULER tokens and improve efficiency.

To further manage computational cost, we introduce RULER tokens at regular intervals $s$ instead of having them for each position:

$$\mathcal{R} = \{r_i : i \in \{0, s, 2s, ..., \lfloor \max(H, W)/s \rfloor \cdot s\}\} \tag{3}$$

In this case, the arithmetic bound is $b = s \times p$. The RULER tokens are generated during the preparation of multimodal inputs. When the input sequence has multiple images, we generate a RULER token sequence before each image with position ID corresponding to each image.

## 3.2 I-MROPE: INTERLEAVED MULTIDIMENSIONAL ROTARY POSITIONAL EMBEDDING

Positional embeddings encode spatial information in vision transformers. Multidimensional RoPE (MRoPE) (Wang et al., 2024b; Bai et al., 2025) extends standard RoPE to VLMs by decomposing positions into multiple spatial-temporal dimensions. However, a critical limitation of MRoPE is that it creates a frequency imbalance between spatial dimensions.

Rotary positional embeddings (RoPE) encode relative positions by applying rotation matrices directly to the query and key vectors in each attention head. Let $m \in \mathbb{N}$ denote the position index of a token and $d$ the dimension of the attention head. For each $2 \times 2$ block, RoPE rotates a pair of dimensions by a position-dependent angle $m\theta_j$. The rotation matrix $\mathbf{R}_{\theta_j,m}$ applied to the query and key vectors is thus expressed as:

$$\mathbf{R}_{\theta_j,m} = \begin{pmatrix} \cos(m\theta_j) & -\sin(m\theta_j) \\ \sin(m\theta_j) & \cos(m\theta_j) \end{pmatrix}, \quad \theta_j = b^{-2j/d}, \tag{4}$$

where $b$ is a hyperparameter called RoPE base. The frequency $\theta_j$ decreases exponentially with the dimension index $j$, producing a spectrum that ranges from high-frequency to low-frequency components as $j$ progresses from 0 to $d$, which is illustrated in the right part of Figure 2. In standard MRoPE, these frequencies are partitioned and assigned consecutively to different spatial-temporal dimensions:

$$\mathbf{R}_{\Theta,t,h,w}^{\mathrm{MRoPE}} = \mathrm{diag}(\mathbf{R}_{\Theta_t,t}, \mathbf{R}_{\Theta_h,h}, \mathbf{R}_{\Theta_w,w}) \tag{5}$$

where $\Theta_t$, $\Theta_h$, and $\Theta_w$ denote disjoint yet consecutive subsets of the frequency spectrum $\theta_j$. This sequential allocation leads to an imbalance: the high-, mid-, and low-frequency parts of the RoPE vector are fully and only occupied by the temporal, height, and width dimensions, respectively. As a result, each dimension is biased towards a limited and different frequency band, constraining the representational capacity and degrading grounding performance across axes (Liu et al., 2024c; Wang et al., 2024c). This imbalance also potentially results in different inner processing mechanisms of each spatial-temporal dimension due to the different modeling behaviors of their corresponding positional embedding.

I-MROPE addresses this imbalance by distributing the frequency spectrum uniformly across spatial dimensions through frequency interleaving. Specifically, instead of assigning consecutive frequency bands to a single axis, each frequency index $j$ is cyclically mapped.

$$\text{Dimension assignment for frequency } j: \quad p_j = \begin{cases} w & \text{if } j \bmod 3 = 0 \\ h & \text{if } j \bmod 3 = 1 \\ t & \text{if } j \bmod 3 = 2 \end{cases} \tag{6}$$

where $p_j$ denotes the spatial dimension (width, height, or temporal) assigned to frequency $\theta_j$.

This interleaving ensures that every dimension receives a full range of frequencies, combining high-frequency components for fine-grained localization with low-frequency components for long-range dependencies. Like vanilla MRoPE, text tokens in the sequence have identical temporal, height, and width indices ($t = h = w = m$), and the formulation reduces exactly to standard RoPE:

$$\mathbf{R}_{\Theta,m,m,m}^{\mathrm{I\text{-}MRoPE}} = \mathbf{R}_{\Theta,m}^{\mathrm{RoPE}} \tag{7}$$

This preserves backward compatibility with pre-trained language models while providing more balanced spatial representations for vision tasks.

## 4 EXPERIMENTAL SETUP

**Training Setup.** We conduct two sets of experiments to validate our approach: training from scratch and finetuning existing VLMs. For the from-scratch experiments, we build on the LLaVA-NeXT framework (Liu et al., 2024b) using SigLIP-SO400M-14@384 (Zhai et al., 2023) as vision encoder and Qwen2.5 7B Instruct (Qwen et al., 2024) as language decoder. We replace the standard 1D positional embeddings in the language decoder in LLaVA-NeXT with MRoPE or I-MRoPE, and integrate RULER tokens into the input sequence during both training and inference.

Following the LLaVA-NeXT training paradigm, we employ a two-stage training process. First, we perform vision-language alignment pretraining on the LLaVA-558K dataset (Liu et al., 2024a),

training only the MLP projection layer. Second, we conduct domain-specific supervised finetuning on UI grounding tasks, training both the projection layer through full finetuning and the language model through LoRA (Hu et al., 2022) for parameter efficiency.

For finetuning experiments, we adapt Qwen2.5-VL 7B Instruct (Bai et al., 2025) by introducing RULER tokens and focus on verifying the significance of RULER alone on grounding performance. We do not change the original model's MRoPE to avoid dramatic changes to the learned model behaviors regarding positional embedding. We use Qwen2.5-VL's default system prompt and chat template for all the finetuning experiments.

In all experiments, we set the RULER token's default interval as $s = 8$ in the main experiments. For I-MRoPE, since GUI grounding does not require a temporal dimension, we use 2D MRoPE and I-MRoPE in the from-scratch training experiments. Specifically, the dimension assignment for frequency $j$ is:

$$\text{Dimension assignment for frequency } j: \quad p_j = \begin{cases} h & \text{if } j \bmod 2 = 0 \\ t & \text{if } j \bmod 2 = 1 \end{cases} \tag{8}$$

The training process follows standard VLM objectives with UI grounding tasks. The model learns to leverage RULER tokens for coordinate prediction while I-MRoPE provides balanced spatial representations throughout the transformer layers. This combination enables precise pixel-level grounding without compromising general vision-language capabilities. More hyperparameter settings can be found in Appendix B.

**Training Data.** Both experimental settings are trained on the UGround dataset (Gou et al., 2025), which provides comprehensive UI grounding annotations on websites. It contains approximately 8M element annotations across 775K screenshots, providing diverse training signals for robust grounding capabilities.

To comply with Qwen2.5-VL's post-training settings regarding coordinates (Bai et al., 2025), we pre-process all coordinates in UGround to use raw pixel values rather than normalized ones. This choice ensures consistency with our RULER token design, which requires each patch's size in terms of the output coordinate to be a square, and avoids the ambiguity introduced by normalization in different aspect ratios.

**Evaluation Setup.** We evaluate our models on three UI grounding benchmarks: ScreenSpot (Cheng et al., 2024), ScreenSpot-V2 (Wu et al., 2025b), and ScreenSpot Pro (Li et al., 2025b). Each benchmark presents screenshots paired with natural language instructions that describe the target UI elements. Models must predict the pixel coordinates corresponding to the described element.

ScreenSpot and ScreenSpot-V2 contain 1,272 instructions each on mobile, desktop, and web platforms, with V2 correcting the annotation errors from the original. ScreenSpot-Pro presents a more challenging scenario with 1,581 tasks from 23 professional desktop-only applications featuring higher resolution interfaces and greater domain shift from typical training data. In particular, ScreenSpot-Pro features higher-resolution images than our training data, making it a strong test of resolution generalization.

We preprocess all benchmarks to use raw pixel coordinates for evaluation, ensuring fair comparison between methods.[2] We measure performance using *Element Accuracy*, which considers a prediction correct if the predicted point falls within the ground-truth bounding box of the target element. We use the evaluation setting and the code provided by Wu et al. (2025a).

**Baselines.** We compare against state-of-the-art UI grounding models of comparable scale. Our baseline models includes Qwen-2-VL 7B Instruct (Wang et al., 2024b), one of the most commonly used open-source VLMs; SeeClick-9.6B (Cheng et al., 2024), an early specialized UI grounding model; OS-Atlas-7B (Wu et al., 2025b), a model designed for operating system interactions; Aguvis-7B (Xu et al., 2025), which uses visual grounding with bounding box supervision; UGround-V1-7B (Gou et al., 2025) trained on the same UGround dataset; UI-TARS-7B (Qin et al., 2025), a recent

---

[2]For baselines trained with normalized coordinates, we apply appropriate transformations to the output to enable comparison.

Table 1: Grounding element accuracy on **ScreenSpot-Pro**. The results of models marked with † are adopted from Wu et al. (2025a). Best results per column within each comparable model group are shown in **bold**. Note that results in the first two groups are not directly comparable to ours, either because the models are closed-source (weights/architectures unavailable) or because their training data and underlying base models are unclear or incomparable. We nevertheless include these numbers for reference.

| Model | Dev | Creative | CAD | Scientific | Office | OS | Avg |
|---|---|---|---|---|---|---|---|
| GPT-4o† | 0.7 | 0.6 | 1.5 | 1.2 | 0.9 | 0.0 | 0.8 |
| Claude Compute† | 12.6 | 16.8 | 11.9 | 25.8 | 26.9 | 8.1 | 17.1 |
| Qwen2-VL-7B† | 1.3 | 0.9 | 0.4 | 3.5 | 3.0 | 0.5 | 1.6 |
| SeeClick-9.6B† | 0.3 | 0.6 | 1.9 | 2.0 | 0.9 | 1.5 | 1.1 |
| OS-Atlas-7B† | 17.7 | 17.9 | 10.3 | 24.4 | 27.4 | 16.8 | 18.9 |
| Aguvis-7B† | 16.1 | 21.4 | 13.8 | 34.6 | 34.3 | 19.4 | 22.9 |
| UGround-V1-7B | 28.1 | 31.7 | 14.6 | 39.0 | 49.6 | 24.5 | 31.1 |
| UI-TARS-7B | 36.1 | 32.8 | 18.0 | 50.0 | 53.5 | 24.5 | 35.7 |
| GUI-Actor-7B + Verifier† | 38.8 | 40.5 | 37.2 | 44.5 | 64.8 | 43.9 | 44.2 |
| *Trained From Scratch with LLaVA-NeXT Framework* | | | | | | | |
| LLaVA-NeXT + LLaVA PE | 23.1 | 25.5 | 12.6 | 35.4 | 43.8 | 20.5 | 26.8 |
| LLaVA-NeXT + MRoPE | 26.8 | 29.4 | 13.6 | 36.5 | 47.5 | 21.2 | 29.2 |
| LLaVA-NeXT + I-MRoPE | 27.1 | 29.8 | 13.8 | 36.6 | 47.8 | 21.5 | 29.4 |
| **LLaVA-NeXT + I-MRoPE + RULER** | **28.2** | **32.1** | **15.3** | **40.5** | **51.6** | **24.8** | **32.1** |
| *Finetuning* | | | | | | | |
| Qwen2.5-VL | 31.4 | 34.2 | 17.1 | 42.8 | 54.0 | 28.3 | 34.6 |
| **Qwen2.5-VL + RULER** | **34.2** | **36.5** | **21.1** | **43.9** | **55.4** | **32.0** | **37.2** |

strong baseline; and GUI-Actor-7B (Wu et al., 2025a) which uses attention-based grounding instead of outputting coordinates. All baseline numbers are reported from original papers or reproduced using official implementations with consistent evaluation protocols. Note that our models use less training data than GUI-Actor. Besides, our models are only trained on UGround and thus have not seen data from domains other than websites, unlike UI-TARS and GUI-Actor.

## 5 RESULTS

### 5.1 GUI GROUNDING PERFORMANCE

We present the comparison among the models trained from scratch with RULER and I-MROPE, the finetuned models equipped with RULER, and the baseline models on ScreenSpot-Pro, ScreenSpot, and ScreenSpot-V2 in Table 1, Table 2, and Table 3, respectively.

For the from-scratch training experiments, multidimensional RoPE consistently outperforms the default 1D RoPE (LLaVA PE) across all benchmarks. Furthermore, our proposed I-MROPE achieves both lower training loss and stronger grounding performance than the original MRoPE, demonstrating the effectiveness of balancing the spectrum across the spatial dimensions. RULER tokens further enhance performance by providing guidance on position-to-coordinate mapping, achieving the best overall results among all models trained from scratch across all datasets. Noticeably, the gains from RULER are most pronounced on ScreenSpot-Pro, reflecting how its reference-then-copy mechanism and bounded pixel coordinate arithmetic across resolutions help generalization to higher resolution grounding tasks.

For fine-tuning experiments, we also observe that adding RULER consistently improves performance, with the largest gains on the higher-resolution ScreenSpot-Pro benchmark. Although RULER does not achieve state-of-the-art results partly due to the limited training data and domains, our experiments nevertheless demonstrate that incorporating RULER reliably enhances grounding performance under comparable training conditions.

Table 2: Grounding element accuracy on **ScreenSpot**. The results of models marked with † are adopted from Wu et al. (2025a). Best results per column within each group are shown in **bold**.

| | M-Text | M-Icon | D-Text | D-Icon | W-Text | W-Icon | Avg |
|---|---|---|---|---|---|---|---|
| GPT-4† | 22.6 | 24.5 | 20.2 | 11.8 | 9.2 | 8.8 | 16.2 |
| GPT-4o† | 20.2 | 24.9 | 21.1 | 23.6 | 12.2 | 7.8 | 18.3 |
| Claude Computer Use† | - | - | - | - | - | - | 83.0 |
| Gemini 2.0† | - | - | - | - | - | - | 84.0 |
| Qwen2-VL-7B† | 75.5 | 60.7 | 76.3 | 54.3 | 35.2 | 25.7 | 55.3 |
| SeeClick-9.6B† | 78.0 | 52.0 | 72.2 | 30.0 | 55.7 | 32.5 | 53.4 |
| OS-Atlas-7B† | 93.0 | 72.9 | 91.8 | 62.9 | 90.9 | 74.3 | 82.5 |
| Aguvis-7B | 95.6† | 77.7 | 93.8 | 67.1 | 88.3 | 75.2 | 84.4 |
| UGround-v1-7B | 93.0 | 79.9 | 93.8 | 76.4 | 90.9 | 84.0 | 86.3 |
| UI-TARS-7B | 94.5 | 85.2 | 95.9 | 85.7 | 90.0 | 83.5 | 89.5 |
| GUI-Actor-7B + Verifier† | 96.0 | 83.0 | 93.8 | 82.1 | 92.2 | 87.4 | 89.7 |
| *Trained From Scratch with LLaVA-NeXT Framework* | | | | | | | |
| LLaVA-NeXT + LLaVA PE | 88.9 | 74.2 | 88.3 | 70.2 | 85.7 | 75.4 | 80.5 |
| LLaVA-NeXT + MRoPE | 90.0 | 76.2 | 90.2 | 72.7 | 88.3 | 77.5 | 82.5 |
| LLaVA-NeXT + I-MRoPE | 90.5 | 76.9 | 90.9 | **73.4** | 88.5 | **77.7** | 83.0 |
| **LLaVA-NeXT + I-MRoPE + Ruler** | **91.4** | **77.0** | **91.5** | 73.2 | **89.5** | 77.2 | **83.3** |
| *Finetuning* | | | | | | | |
| Qwen2.5-VL | 93.4 | 80.5 | **94.6** | 76.4 | 91.1 | 84.6 | 86.8 |
| **Qwen2.5-VL + Ruler** | **94.2** | **84.1** | 93.6 | **76.5** | **92.4** | **85.3** | **87.7** |

Table 3: Grounding element accuracy on **ScreenSpot-V2**. The results of models marked with † are adopted from Wu et al. (2025a). Best results per column within each group are shown in **bold**.

| | M-Text | M-Icon | D-Text | D-Icon | W-Text | W-Icon | Avg |
|---|---|---|---|---|---|---|---|
| GPT-4o + OmniParser-v2† | 95.5 | 74.6 | 92.3 | 60.9 | 88.0 | 59.6 | 80.7 |
| SeeClick-9.6B† | 78.4 | 50.7 | 70.1 | 29.3 | 55.2 | 32.5 | 55.1 |
| OS-Atlas-7B† | 95.2 | 75.8 | 90.7 | 63.6 | 90.6 | 77.3 | 84.1 |
| Aguvis-7B† | 95.5 | 77.3 | 95.4 | 77.9 | 91.0 | 72.4 | 86.0 |
| UGround-V1-7B | 95.0 | 83.3 | 95.0 | 77.8 | 92.1 | 77.2 | 87.6 |
| UI-TARS-7B | 96.9 | 89.1 | 95.4 | 85.0 | 93.6 | 85.2 | 91.6 |
| GUI-Actor-7B + Verifier† | 97.2 | 84.8 | 94.3 | 85.0 | 94.0 | 85.2 | 90.9 |
| *Trained From Scratch with LLaVA-NeXT Framework* | | | | | | | |
| LLaVA-NeXT + LLaVA PE | 92.4 | 78.8 | 90.1 | 75.3 | 87.9 | 74.1 | 83.1 |
| LLaVA-NeXT + MRoPE | 93.2 | 79.1 | 90.8 | 76.6 | 88.0 | 76.3 | 84.0 |
| LLaVA-NeXT + I-MRoPE | 93.4 | 80.0 | **91.3** | 77.5 | 88.1 | 76.7 | 84.5 |
| **LLaVA-NeXT + I-MRoPE + Ruler** | **95.0** | **82.7** | 90.3 | **79.8** | **88.6** | **77.1** | **85.6** |
| *Finetuning* | | | | | | | |
| Qwen2.5-VL | 95.6 | 85.2 | 95.2 | **80.8** | 92.5 | 79.9 | 88.2 |
| **Qwen2.5-VL + Ruler** | **96.2** | **87.0** | **95.3** | 80.5 | **93.2** | **81.6** | **89.0** |

## 5.2 Analysis on Ruler Token Interval

To analyze the effect of changing the interval of the Ruler token, we provide a sensitivity analysis of $s$ in Equation 3. The results are shown in Figure 3.

In the figure, we notice that all interval settings yield consistent improvements compared to models without Ruler tokens in all datasets. However, varying the Ruler token interval does not yield significant or consistent improvements on the benchmarks. Based on the results, we adopt the setting of $s = 8$ as a good trade-off between performance and efficiency. However, it should be noted that in extremely low-resolution settings such as mobile phone screenshot grounding, an interval $s = 16$ may inject only a single Ruler token, leading to reduced performance in the mobile-related subtasks of ScreenSpot and ScreenSpot-V2.

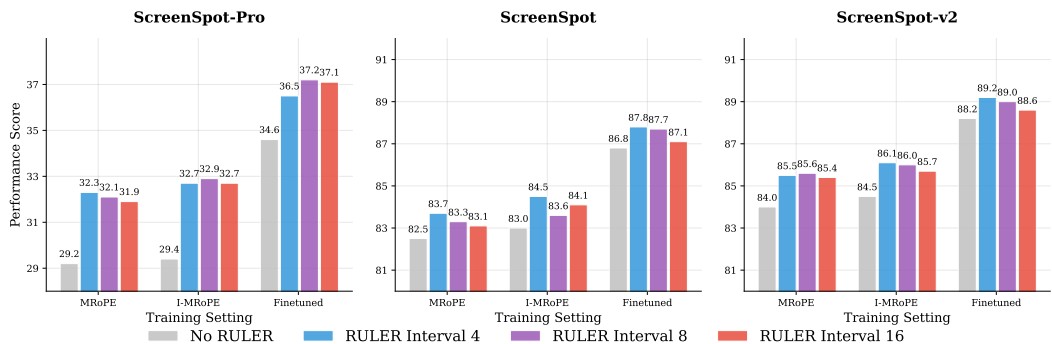

Figure 3: Ablation study on RULER token intervals $s$ across different benchmarks and training settings.

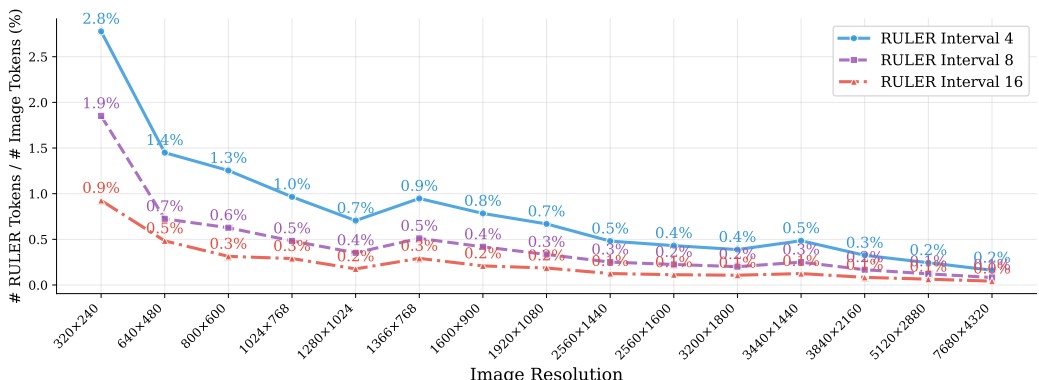

Figure 4: Analysis of the ratio of the number of RULER tokens to the number of image tokens under common mobile phone and computer screen resolutions for different RULER intervals. All numbers are in percentages (%).

## 5.3 EFFICIENCY ANALYSIS

To demonstrate the efficiency of adding RULER tokens, we provide an efficiency analysis in the $p = 8$ setting in Figure 4. In this figure, we report the ratio of RULER tokens to image tokens in common resolutions of mobile phones and computer screens under different interval settings. Even in the extreme 8K screenshot scenarios and using an interval of $s = 2$, RULER only adds 68 additional tokens, which is merely 0.2% of the total number of vision tokens. For low-resolution mobile screenshots, the highest ratio of RULER to vision tokens observed is 2.8%, where the impact on efficiency remains negligible. These results confirm that the introduction of tokens RULER can effectively improve grounding performance while maintaining efficiency.

## 6 CONCLUSIONS AND LIMITATIONS

We presented a framework for GUI grounding that replaces implicit position-to-pixel coordinate mapping with explicit spatial guidance. RULER tokens provide coordinate references that transform unstable regression into robust reference and adjustment, while I-MRoPE corrects frequency imbalances in the positional embeddings. Our approach achieves consistent improvements across benchmarks, with particularly strong gains on high-resolution displays beyond training resolutions, validating its generalization capability. The minimal computational overhead (less than 1% of token increase) makes deployment practical. Future work could explore adaptive token placement and extension to video interfaces. The success of explicit spatial guidance over implicit learning suggests broader applications beyond GUI automation for any task that requires precise visual localization.

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

## A LLM USAGE STATEMENT

For this work, GitHub Copilot provided light coding assistance during human-authored code development. Claude was used for grammar check and language polishing of manually written text sections after completion.

## B IMPLEMENTATION DETAILS

We provide detailed training configurations for our experiments in the following. All experiments are performed on 8 NVIDIA H100 GPUs.

### B.1 TRAINING FROM SCRATCH

**Stage 1: Vision-Language Alignment Pretraining.** We follow the LLaVA-NeXT training paradigm. The model uses SigLIP-SO400M-14@384 (Zhai et al., 2023) as the vision encoder and Qwen2.5 7B Instruct (Qwen et al., 2024) as the language model. During pretraining, we train only the MLP projection layer while keeping both vision and language models frozen. Training is performed on the LLaVA-558K dataset (Liu et al., 2023) for 1 epoch with a learning rate of $1 \times 10^{-3}$ using cosine scheduling and 3% warmup ratio. We use a per-device batch size of 4 with gradient accumulation steps of 2, resulting in an effective batch size of 64 across 8 GPUs. The maximum sequence length is set to 8,192 tokens. Images are processed using the AnyRes configuration with a maximum of 9 patches and grid pinpoints ranging from $(1 \times 1)$ to $(12 \times 6)$ to accommodate high-resolution images during inference. We employ DeepSpeed Zero-2 with CPU offload (Ren et al., 2021) and mixed precision training (bf16) for memory efficiency. For models using RULER, we set the token interval to $s = 8$, while positional embedding configurations (default LLaVA PE, MRoPE, or I-MRoPE) are specified throughout the pretraining and finetuning process.

**Stage 2: Domain-Specific Finetuning.** Using the pretrained projection layer from Stage 1, we finetune on the UGround dataset (Gou et al., 2025) with coordinates converted to raw pixel values to match our RULER token design. In this stage, we train the projection layer with full parameter finetuning and the language model using LoRA (Hu et al., 2022) with rank 16 for parameter efficiency. The base learning rate is set to $1 \times 10^{-5}$ for the projection layer and LoRA parameters. We

use a per-device batch size of 1 with gradient accumulation steps of 4, yielding an effective batch size of 32. The maximum sequence length is extended to 16,384 tokens to accommodate higher-resolution images. Training runs for 1 epoch with cosine learning rate scheduling and 3% warmup. We continue using DeepSpeed Zero-2 with CPU offload and bf16 mixed precision.

## B.2 FINETUNING QWEN2.5-VL

For adapting the pretrained Qwen2.5-VL 7B Instruct model (Bai et al., 2025), we use a conservative finetuning approach to preserve the existing capabilities of the model while adding RULER tokens. We maintain the model's original MRoPE configuration to avoid disrupting learned position-aware behaviors. The model is finetuned with a low learning rate of $1 \times 10^{-5}$ using cosine scheduling with 3% warmup to ensure stable adaptation. We use a per-device batch size of 4 with gradient accumulation steps of 4, resulting in an effective batch size of 128. The maximum sequence length remains at 16,384 tokens, and we utilize Qwen2.5-VL's dynamic resolution capability with pixel counts ranging from 784 to 50,176. Training runs for 1 epoch on the UGround dataset with all components (vision encoder, MLP projector, and language model) being trainable. We employ DeepSpeed Zero-3 (Rajbhandari et al., 2019) for distributed training and bf16 mixed precision. RULER tokens are integrated into the input sequence with interval $s = 8$ when specified, and we use Qwen2.5-VL's native chat template and system prompts for consistency with the pretrained model's behavior.

## B.3 EVALUATION PROTOCOL

All models are evaluated using greedy decoding (temperature=0) with the same maximum sequence length as training. For ScreenSpot benchmarks, we preprocess all coordinates to raw pixel values and use the evaluation code from Wu et al. (2025a). Element accuracy is computed by checking if the predicted coordinate falls within the ground-truth bounding box. We ensure consistent preprocessing across all baselines for fair comparison.

