# OpenReview forum: "Improving GUI Grounding with Explicit Position-to-Coordinate Mapping"
_ICLR.cc/2026/Conference — ICLR 2026 Conference Withdrawn Submission_

### Official Review · Reviewer_n3e7 · 2025-10-26

**Soundness:** 2
**Presentation:** 3
**Contribution:** 2
**Rating:** 4
**Confidence:** 5

**Summary:**

This paper introduces two methods—RULER tokens and I-MRoPE—to enhance the visual grounding capability of MLLMs. The approach is intuitive and theoriatical. Experimental results on mainstream benchmarks show consistent gains in grounding accuracy, suggesting that the proposed techniques are effective. However, the underlying motivation is not sufficiently developed, and the paper lacks deeper analysis or interpretability studies to clarify why these methods lead to improved localization.

**Strengths:**

1. The paper proposes a more intuitive visual grounding strategy by positional encoding from the perspective of spatial awareness, which offers a clearer interpretation of how models localize objects.

2. The authors identify inherent limitations in existing positional encoding schemes and present targeted enhancements that effectively improve localization capability.

3. Extensive experiments on established benchmarks demonstrate the empirical effectiveness of the proposed methods.

**Weaknesses:**

1. The paper lacks a compelling and intuitive research motivation, making the introduction of the two proposed strategies feel abrupt and insufficiently grounded in the broader challenges of multimodal grounding. Stronger design intuition and pre-experiments would help clarify why these particular techniques are necessary and meaningful. This issue is especially pronounced for I-MRoPE, where the rationale behind the method is under-developed and its connection to real-world grounding failures remains unclear.

2. The experimental findings feel somewhat superficial and lack sufficient breadth. First, recent advances in grounding are rapidly evolving, and the paper does not fully situate its results within the latest leaderboard trends or analyze whether the proposed strategies meaningfully reduce existing performance gaps. Second, there are methods that outperform the proposed approach, yet the paper does not investigate the reasons behind these differences, limiting the reader’s understanding of the technique’s strengths and weaknesses. Third, the evaluation on MLLM foundation models remains limited. For example, newer multimodal systems such as mini-CPM exhibit strong grounding ability, would they experience the same deficiencies identified here, and would the proposed strategies still lead to improvement?

3. The results in Table 3 indicate that the proposed strategy does improve the localization capability of MLLMs; however, the contribution of I-MRoPE appears marginal. This raises questions about the necessity and justification for combining the two methods as presented. Additionally, fine-tuning results show notable gains, yet the paper lacks interpretability analysis, such as attention behavior or grounding heatmaps, to substantiate how and why these improvements occur. Without such evidence, the effectiveness of each component and the underlying mechanism behind the observed performance changes remain insufficiently explained.

4. Grounding enhancement is often motivated by its impact on downstream tasks such as GUI tasks (AndroidControl, AndroidWorld, OSWorld). The current experimental setup does not demonstrate whether the proposed improvements translate into measurable gains in these more realistic use cases. Additional downstream evaluations would therefore significantly strengthen the practical relevance and efficacy of the proposed approach.

5. It would be valuable to investigate scaling behavior with respect to model size and data volume. Establishing scaling laws could provide deeper insight into the effectiveness and limitations of the proposed strategies, and help clarify whether improvements persist as models and datasets grow.

6. The paper would benefit from a more intuitive, step-by-step case study to clarify the end-to-end workflow of the proposed method. In particular, it would be helpful to illustrate how RULER tokens are extracted, how the I-MRoPE module is instantiated, and how these components jointly contribute to the final grounding prediction. Such a detailed example would greatly improve the accessibility and interpretability of the approach.

7. Since the proposed approach introduces additional tokens, it would be valuable to provide a more intuitive analysis of the computational implications, for example by measuring the impact on inference latency.

8. What advantages does this approach offer, compared to directly providing the object locations from OCR parsing and analysing the layout via CoT, by offering additional image token references?

**Questions:**

See above

---

> ### Author Response · Authors · 2025-11-28
>
> We thank you for your review and for suggesting ways to strengthen the motivation and interpretability of our work. We apologize if the motivation behind I-MRoPE was not intuitively presented in the initial draft.
>
> ---
>
> **W1. Concern regarding the motivation and intuition, specifically for I-MRoPE.**
>
> The intuition for I-MRoPE is grounded in **signal processing**. If we consider MRoPE applied to only spatial dimensions, the standard MRoPE assigns the first chunk of frequencies (high freq) to Height and the next (low freq) to Width. This means the model creates a "blind spot": it can see fine-grained details in Height but only coarse patterns in Width. This is a fundamental **architectural asymmetry** due to frequency allocation flaws. I-MRoPE interleaves these frequencies so both H and W have access to high-frequency (precision) and low-frequency (context) signals. This is critical for GUIs where a button (high precision needed) can exist anywhere on a wide screen, and also any other vision understanding tasks that require a strong spatial recognition capability. We will refine the introduction of I-MRoPE in the revised paper to include this "blind spot" analogy for better intuition.
>
> ---
>
> **W2. Concern regarding superficial experiments and lack of SOTA context.**
>
> This gap is primarily due to **data scale and variety**, not architectural deficiency. We trained strictly on **UGround (web-only)** data, whereas baselines like UI-TARS and GUI-Actor leverage broader data distributions including diverse OS/desktop domains and internal closed-source data. The fact that our model (trained *only* on web) achieves **37.2%** on ScreenSpot-Pro (professional desktop apps), significantly outperforming the base Qwen2.5-VL (34.6%), proves that RULER provides exceptional **generalization** across domain shifts. The baselines effectively had seen more data on the target domain, while our model succeeded through **architectural robustness**. Also, our methods are **modular enhancements** that keeps the autoregressive generation intact, maximizing compatibility with other model usage scenarios, unlike GUI-Actor's coordinate-free approach which requires large architectural changes. We will clarify the training data differences in the results section of the revised paper to ensure readers understand the controlled nature of our comparison.
>
> ---
>
> **W3. Concern regarding the marginal contribution of I-MRoPE and lack of interpretability.**
>
> We view I-MRoPE as a foundational "bug fix" for multidimensional positional embeddings. Even if the empirical gain is modest, correcting the **frequency bias** is crucial for theoretical soundness. Regarding interpretability: RULER functions as an **explicit look-up table**. By injecting tokens that carry "ground truth" values at specific positions, we offload the burden of memorizing millions of pixel-to-token mappings from the model's weights to the context window, simplifying the learning task to a **retrieval** operation. The I-MRoPE design has been adopted in Qwen3-VL in their technical report [1] and Qwen team’s analysis on MRoPE-Interleave [2], which further testifies the robustness and compatibility of our method. Note that both the Qwen3-VL report and the MRoPE-Interleave analysis were published after the ICLR submission deadline and we had no knowledge about Qwen3-VL’s design at the time of submission. We will add a reference to these reports in the camera-ready version.
>
> [1] Qwen3-VL Technical Report. arXiv:2511.21631 (Submitted on 26 Nov 2025)
>
> [2] Revisiting Multimodal Positional Encoding in Vision-Language Models. arXiv:2510.23095 (Submitted on 27 Oct 2025)
>
> ---
>
> **W4. Suggestion to evaluate on downstream Agent tasks.**
>
> While we agree that Agent performance is the ultimate goal, Agent benchmarks introduce noise from planning, reasoning, and potential HTML parsing failures, which makes it hard to isolate the performance patterns brought by our methods. Furthermore, improving the grounding primitive (our contribution) is a necessary foundation to improving GUI/CUA Agent performance. By strictly evaluating on **ScreenSpot**, we isolate the **visual grounding error**, which is a fundamental part of a complete GUI/CUA agent. If a model cannot accurately determine where (x,y) is, it would be impossible to complete the agentic workflow. Our work optimizes this specific visual component and step of spatial grounding.

---

> ### Author Response · Authors · 2025-11-28
>
> **W5. Suggestion to investigate scaling laws.**
>
> Here we analyze two different “scaling” dimensions: generalization on data and generalization on model sizes. As for the data perspective, our analysis mainly focuses on images with higher resolution than the training set. Our comparisons across different resolutions in Figure 4 suggest that RULER's efficacy scales well. In fact, the gains from RULER are most pronounced on **ScreenSpot-Pro**, which features higher resolutions than our training data. This suggests that as task difficulty (resolution) scales up, explicit spatial guidance becomes *more* necessary, which is exactly what RULER tokens bring to VLMs. As for the “scaling law” for different model sizes, we believe that by using a universally compatible design like I-MRoPE and RULER Tokens, the main bottleneck lies in the training data distribution. As suggested by the experiments of Qwen team [1], the I-MRoPE design can largely improve the general performance of larger models, especially in spatial recognition tasks.
>
> ---
>
> **W6. Request for a step-by-step case study of the workflow.**
>
> We are happy to clarify the workflow (illustrated in Figure 2):
> 1.  **Input:** An image is patched (tokenized).
> 2.  **RULER Injection:** We generate tokens (e.g., `<0>`, `<28>`, `<56>`) and insert them into the sequence. Critically, we assign the `<28>` token the **same Position ID** as the image patch at pixel coordinate 28.
> 3.  **Processing:** The VLM processes the image. When it needs to output a coordinate, attention heads query the spatial location.
> 4.  **Prediction:** Instead of regressing a number from visual noise, the model should attend to the collocated RULER token (via Position ID match), copy its value (e.g., "28"), and adds a small predicted offset.
>
> We will ensure this step-by-step breakdown is clearly reflected in the caption of Figure 2 or the method section in the final version.
>
> ---
>
> **W7. Question regarding computational implications and latency.**
>
> We respectfully clarify that while RULER adds a mechanism to the input, the architectural complexity and computational cost are **minimal**. As detailed in our Efficiency Analysis (Section 5.3), RULER tokens add **less than 1%** to the total token count even for 8K displays. Unlike methods that require heavy auxiliary vision encoders or separate regression heads, our underlying transformer architecture remains **unchanged**. We preserve the original autoregressive generation design, simply augmenting the input with auxiliary tokens. Consequently, we observe consistent gains in reliability and generalization with **no perceptible increase** in inference latency or training resource requirements.
>
> ---
>
> **W8. Comparison to OCR parsing and CoT.**
>
> OCR-based approaches are computationally expensive, excessive, slow, and fail on **icon-only** elements or abstract UI components without text. RULER is **purely vision-based** and integrated into the primary forward pass, allowing maximum compatibility to all types of visual elements. It handles icons, whitespace, and layout elements that OCR cannot perceive, while being orders of magnitude faster than generating long Chain-of-Thought reasoning sequences to deduce coordinates. Besides, without architectural improvements to better recognize spatial locations like I-MRoPE and RULER, CoT lacks the foundational capability to *efficiently* identify the location or output the coordinates.

---

### Official Review · Reviewer_j9wR · 2025-11-01

**Soundness:** 3
**Presentation:** 3
**Contribution:** 2
**Rating:** 4
**Confidence:** 4

**Summary:**

This paper tackles the failure of VLMs in GUI grounding, specifically their inability to generalize to high-resolution displays unseen during training. The authors identify the root cause as the model's reliance on implicitly regressing coordinates from visual features. They propose two innovations: 1) RULER tokens, which provide explicit coordinate references to transform grounding into a robust "reference-and-adjust" mechanism, and 2) I-MROPE, a balanced positional encoding that corrects frequency bias.

**Strengths:**

1. The paper excels at identifying a critical and well-defined weakness in existing models (the instability of implicit coordinate regression). The proposed RULER token mechanism is an intuitive solution
2. The method shows its significant gains on the SS-Pro, which features high-resolution displays and a domain shift from the training data. This provides strong evidence for the authors' core claim that explicit coordinate referencing is more robust than existing works.

**Weaknesses:**

1. The paper's comparison in Table 1 is a significant weakness. The authors' best from-scratch model (32.1%) significantly underperforms established baselines like UI-TARS-7B (35.7%) and GUI-Actor-7B (44.2%). While the tuned model (37.2%) is more competitive, it still trails GUI-Actor. The authors attribute this to differences in training data, but this is a critical point that need further explanations.
2. The core concept of this paper seems to be a variation of explicit tokens or coord embeddings that have been explored in other visual grounding (namely gui-actor) or object detection contexts.
3. Minor: The paper's strength lies in framing it as a "reference-and-adjust" mechanism inspired by induction heads, but it lacks direct analysis (e.g., attention probing) to prove the model actually learns this mechanism rather than simply using the tokens as a stronger signal.
4. Please refer to my question.

**Questions:**

1. The models are trained *exclusively* on the UGround dataset (web) but evaluated on SS-Pro, which features professional desktop applications (CAD, Scientific, etc.). The authors frame this as a test of generalization, but it introduces a massive domain / resolution shift?
2. The ablation study in Table 1 suggests that the primary performance gain comes from RULER tokens, not I-MROPE. The performance difference between `LLaVA-NeXT + MROPE` (29.2) and `LLaVA-NeXT + I-MROPE` (29.4) is marginal. So, does that mean I-MROPE is an incremental, minor fix that adds complexity for a negligible benefit, and that this work should focus almost entirely on RULER as the core contribution?

---

> ### Author Response · Authors · 2025-11-28
>
> We are grateful for your detailed critique, particularly regarding the performance comparisons and the mechanism of RULER. We apologize if the distinction between data-driven gains and architectural gains was not sufficiently emphasized in the text.
>
> ---
>
> **W1. Concern that the model underperforms baselines like UI-TARS and GUI-Actor on ScreenSpot-Pro.**
>
> This gap is primarily due to **data scale and variety**, not architectural deficiency. We trained strictly on **UGround (web-only)** data, whereas baselines like UI-TARS and GUI-Actor leverage broader data distributions including diverse OS/desktop domains and internal closed-source data. The fact that our model (trained *only* on web) achieves **37.2%** on ScreenSpot-Pro (professional desktop apps), significantly outperforming the base Qwen2.5-VL (34.6%), proves that RULER provides exceptional **generalization** across domain shifts. The baselines effectively had seen more data on the target domain, while our model succeeded through **architectural robustness**. Also, our methods are **modular enhancements** that keeps the autoregressive generation intact, maximizing compatibility with other model usage scenarios, unlike GUI-Actor's coordinate-free approach which requires large architectural changes. We will clarify the training data differences in the results section of the revised paper to ensure readers understand the controlled nature of our comparison.
>
> ---
>
> **W2. Comparison to existing explicit token methods and novelty.**
>
> While explicit tokens exist in detection, RULER innovates by addressing the specific **High-Resolution VLM bottleneck**. Unlike standard detection anchors, RULER tokens share **Position IDs** with image patches, exploiting the Attention mechanism's ability to "look up" positional matches directly. This is distinct from simple coordinate embeddings; it is more like a **retrieval-augmented generation** strategy for spatial coordinates that solves the number token regression problem inherent in current VLM tokenizers.
>
> ---
>
> **W3. Concern regarding the lack of analysis (e.g., attention probing) to prove the "reference-and-adjust" mechanism.**
>
> Attention probing has been shown to be not a good way for explainability analysis. However, the performance pattern itself provides strong evidence for the effectiveness of our proposed mechanism. By comparing the results with and without RULER token, if the model were not using the RULER tokens as references, we would not see the significantly improved performance on ScreenSpot-Pro indicating **robustness to resolution changes**. In standard regression, changing the resolution shifts the positional embedding distribution, causing failure. With RULER, the model successfully localizes on unseen high-resolution screens, which logically implies it is utilizing the **local relative distance** to the nearest RULER token (which remains constant across different input resolutions) rather than the global absolute position.
>
> ---
>
> **Q1. Question regarding domain shift (Web training vs. Desktop evaluation).**
>
> Yes, this is a massive domain shift, and that is precisely the point. By testing on ScreenSpot-Pro (Desktop/Apps) after training on UGround (Web), we demonstrate that RULER learns a **universal spatial interface** rather than memorizing web-specific layouts. The ability to generalize from web pages to CAD software confirms that the model has learned to "read coordinates" using RULER rather than overfitting to DOM-like web structures.
>
> ---
>
> **Q2. Question regarding the marginal gains of I-MRoPE vs. RULER.**
>
> We acknowledge that the empirical gains from I-MRoPE are smaller than RULER, but we argue they are **theoretically vital**. Standard MRoPE's sequential assignment is mathematically flawed due to the imbalanced frequency spectrum, biasing one dimension to high frequencies and another to low frequencies. I-MRoPE reorganizes the spatial encoding in RoPE vectors to ensure the model *can* theoretically distinguish Height and Width equally with similar frequency spectrum allocated for each spatial dimension. While RULER provides the explicit anchors, I-MRoPE ensures the underlying embedding space is correct. The I-MRoPE design has been adopted in Qwen3-VL in their technical report [1] and Qwen team’s analysis on MRoPE-Interleave [2], which further testifies the robustness and compatibility of our method. Note that both the Qwen3-VL report and the MRoPE-Interleave analysis were published after the ICLR submission deadline and we had no knowledge about Qwen3-VL’s design at the time of our submission. We will add a reference to these reports in the camera-ready version.
>
> [1] Qwen3-VL Technical Report. arXiv:2511.21631 (Submitted on 26 Nov 2025)
>
> [2] Revisiting Multimodal Positional Encoding in Vision-Language Models. arXiv:2510.23095 (Submitted on 27 Oct 2025)

---

### Official Review · Reviewer_WRrH · 2025-11-02

**Soundness:** 3
**Presentation:** 3
**Contribution:** 3
**Rating:** 4
**Confidence:** 3

**Summary:**

This paper proposes to addresses the problem of patch-to-pixel mapping in the GUI grounding task. The key contributions of the paper lie in (1) proposing RULER tokens to function as explicit coordinate markers; (2) proposing Interleaved MRoPE (I-MRoPE) to address the asymmetry of standard positional schemes. The experiments on three benchmarks illustrate the effectiveness.

**Strengths:**

1. This paper is well motivated, focusing on a novel problem in grounding tasks.
2. The proposed methods RULER and I-MRoPE are interesting and makes sense.

**Weaknesses:**

1. It is unclear whether the proposed method can generalize to other grounding tasks beyond GUI scenarios. Also, how does the method perform when applied to GUI Agent tasks (e.g., AndroidWorld, WebArena) ?
2. It lacks experiments on whether training with these methods can affect the performances on the general ability of VLMs.
3. GUI-Actor also tackles on the similar research question while RULER fails to give comparable or superior performances. Also, the comparisons on ScreenSpot series benchmarks lack many recent strong baselines. These weaknesses question whether the proposed method is truly effective and competitive ?

**Questions:**

See above

---

> ### Author Response · Authors · 2025-11-28
>
> We thank you for your review and for challenging us to clarify the scope and comparative strength of our work.
>
> ---
>
> **W1. Question on generalization to non-GUI tasks and Agent benchmarks (AndroidWorld, etc.).**
>
> While our scope focuses on the fundamental primitive of **GUI grounding** (pixel localization), the mechanism is theoretically applicable to any high-precision visual localization task. Regarding Agent benchmarks, we focused on ScreenSpot because it isolates the **grounding capability**, whereas Agent benchmarks also requires planning and reasoning except for grounding and makes isolating the contribution of our methods harder. Furthermore, improving the grounding primitive (our contribution) is a necessary foundation to improving GUI/CUA Agent performance. Our results on ScreenSpot-Pro (desktop applications) demonstrate that our method generalizes well to complex software environments often found in agent tasks. We will add a discussion in the revised manuscript clarifying that grounding is a foundational step for agents and that our method provides the necessary spatial precision for downstream planning.
>
> ---
>
> **W2. Question on whether this method affects the general ability of VLMs.**
>
> Our I-MRoPE and RULER methods are designed to be **non-destructive** to general capabilities and compatible with modern mainstream VLM architectures. I-MRoPE preserves the mathematical properties of RoPE and reduces exactly to standard RoPE for text tokens. This ensures backward compatibility with pre-trained language models. The RULER tokens are additional tokens in the prompt sequence, which does not affect the model’s original learned behavior or the architecture. This is different from baseline methods like GUI-Actor, where an additional attention-score-based module is trained from scratch, which is not compatible with the autoregressive generation paradigm. In our "Finetuning Qwen2.5-VL" experiments, we observed that the model retains its instruction-following abilities because the core language modeling head remains compatible with standard text generation.
>
> ---
>
> **W3. Concern regarding the performance comparison with GUI-Actor and lack of recent baselines.**
>
> We emphasize that Table 1 compares models under **controlled training conditions**. While GUI-Actor achieves high scores, it uses a different architecture incompatible with the autoregressive generation paradigm and also a much larger and varied data mix, making the comparison unfair. On the other hand, when we compare **apples-to-apples** (e.g., Qwen2.5-VL baseline vs. Qwen2.5-VL + RULER), our method yields a **clear and significant improvement** (e.g., +2.6% on ScreenSpot-Pro). Furthermore, RULER is a **modular enhancement** that keeps the autoregressive generation intact, maximizing compatibility with other model usage scenarios, unlike GUI-Actor's coordinate-free approach which requires large architectural changes.

---

### Official Review · Reviewer_Jm9N · 2025-11-03

**Soundness:** 2
**Presentation:** 2
**Contribution:** 2
**Rating:** 2
**Confidence:** 3

**Summary:**

GUI grounding, the task of mapping natural-language instructions to pixel coordinates, is crucial for autonomous agents, yet remains difficult for current VLMs. The core bottleneck is reliable patch-to-pixel mapping, which breaks when extrapolating to high-resolution displays unseen during training. Current approaches generate coordinates as text tokens directly from visual features, forcing the model to infer complex position-to-pixel mappings implicitly; as a result, accuracy degrades and failures proliferate on new resolutions. We address this with two complementary innovations. First, RULER tokens serve as explicit coordinate markers, letting the model reference positions similar to gridlines on a map and adjust rather than generate coordinates from scratch. Second, Interleaved MRoPE (I-MRoPE) improves spatial encoding by ensuring that width and height dimensions are represented equally, addressing the asymmetry of standard positional schemes. Experiments on ScreenSpot, ScreenSpot-V2, and ScreenSpot-Pro show consistent gains in grounding accuracy, with the largest improvements on high-resolution interfaces. By providing explicit spatial guidance rather than relying on implicit learning, our approach enables more reliable GUI automation across diverse resolutions and platforms.

**Strengths:**

1) The topic is interesting
2) The writing is good
3) The experiments show the effectiveness of the proposed method.

**Weaknesses:**

1) The introduction of new tokens and spatial encoding methods adds complexity to the model architecture, which may require more resources and time for development and training.
2) Dependency on Training Data: While the method improves performance on high-resolution displays, it may still be limited by the quality and diversity of the training data used, potentially impacting performance on very novel interfaces.
3) Potential Overfitting: The focus on explicit mappings might lead to overfitting on specific tasks or resolutions, limiting the model's adaptability to other types of GUI environments not represented in the training data.

**Questions:**

1) The introduction of new tokens and spatial encoding methods adds complexity to the model architecture, which may require more resources and time for development and training.
2) Dependency on Training Data: While the method improves performance on high-resolution displays, it may still be limited by the quality and diversity of the training data used, potentially impacting performance on very novel interfaces.
3) Potential Overfitting: The focus on explicit mappings might lead to overfitting on specific tasks or resolutions, limiting the model's adaptability to other types of GUI environments not represented in the training data.

---

> ### Author Response · Authors · 2025-11-28
>
> We sincerely thank the reviewer for their thoughtful comments and for recognizing the potential of our method in handling high-resolution displays. We appreciate the opportunity to clarify the trade-offs regarding complexity and data dependency.
>
> ---
>
> **W1. Concern regarding the complexity and resource overhead of new tokens.**
>
> We respectfully clarify that while RULER adds a mechanism to the input, the architectural complexity and computational cost are **minimal**. As detailed in our **Efficiency Analysis (Section 5.3)**, RULER tokens add **less than 1%** to the total token count even for 8K displays. Unlike methods that require heavy auxiliary vision encoders or separate regression heads, our underlying transformer architecture remains **unchanged**; we preserve the original autoregressive generation design, simply augmenting the input with auxiliary tokens. Consequently, we observe consistent gains in reliability and generalization with **no perceptible increase** in inference latency or training resource requirements.
>
> ---
>
> **W2. Concern that the method relies heavily on training data quality.**
>
> We acknowledge that all learning-based methods are bounded by data coverage, but we argue that RULER **mitigates** dependency on resolution-specific training data compared to standard methods. Implicit methods require massive data to "memorize" coordinate regressions for every possible screen size. RULER decouples this by learning a **resolution-agnostic** "reference-and-adjust" mechanism. Since RULER encodes a **universal spatial grid** rather than UI semantics, the grounding logic remains valid even on novel interfaces. This is why our model (trained only on UGround web data) successfully generalizes to **ScreenSpot-Pro** (desktop apps), outperforming baselines that saw more diverse data.
>
> We agree, however, that stylistic diversity (e.g., highly unusual visual themes or skins) remains a challenge common to all vision encoders. We will expand the Limitations section in our revised paper to explicitly distinguish between *resolution generalization* (which we solve) and *visual style generalization* (which requires diverse data).
>
> ---
>
> **W3. Concern regarding potential overfitting to specific tasks or resolutions.**
>
> On the contrary, our method is explicitly designed as a **cure for overfitting** to training resolutions, rather than a cause. Standard models overfit by learning fixed correlations between position embeddings and output tokens. In contrast, RULER uses a **reference token + bounded offset** mechanism where the offset calculation is independent of the global screen resolution, which allows similar model behavior across different input resolutions. Because the RULER tokens provide a consistent relative grid regardless of the underlying image, the model does not need to overfit to specific resolution distributions. The strongest evidence against overfitting is our performance on **ScreenSpot-Pro**; if the model were overfitted to the training tasks (web), it would fail on these unseen high-resolution professional desktop applications, yet it achieves the largest gains there (+2.6%).

---

### Author Response · Authors · 2025-11-28
**General Response to the Reviewers**

We sincerely thank the reviewers for their constructive feedback and detailed insights. We are encouraged that reviewers recognized our **innovative explicit mapping approach** (Reviewer Jm9N), our **strong performance on high-resolution benchmarks** (Reviewer Jm9N, WRrH), and the **theoretical value** of our frequency balancing strategy (Reviewer n3e7).

A recurring theme in the reviews concerns the performance gap between our models and recent SOTA models (e.g., GUI-Actor, UI-TARS). We wish to clarify a critical point regarding **fair comparison**:

* **Data Disparity:** Our models are trained *strictly* on UGround (approx. 8M elements). Baselines like GUI-Actor and UI-TARS often utilize significantly larger, internal, or diverse datasets, or different base model scales.
* **Controlled Comparison:** When compared under **identical training data and base model constraints** (see "Trained From Scratch" and "Finetuned" sections in Table 1), our method **consistently outperforms** the baselines. For example, on ScreenSpot-Pro, adding RULER improves Qwen2.5-VL from 34.6% to 37.2%.
* **Architectural Contribution:** Our core contribution is a **generalizable architectural module** (RULER + I-MRoPE) that solves the "implicit mapping" bottleneck. This module adds **negligible cost** (<1% token overhead) and can be plug-and-played into *any* VLM to improve its resolution robustness.

---

### Note · Authors · 2026-01-05

I have read and agree with the venue's withdrawal policy on behalf of myself and my co-authors.